# The Influence of Nutrition and Physical Activity on Exercise Performance after Mild COVID-19 Infection in Endurance Athletes-CESAR Study

**DOI:** 10.3390/nu14245381

**Published:** 2022-12-18

**Authors:** Daniel Śliż, Szczepan Wiecha, Jakub S. Gąsior, Przemysław Seweryn Kasiak, Katarzyna Ulaszewska, Marek Postuła, Łukasz A. Małek, Artur Mamcarz

**Affiliations:** 13rd Department of Internal Medicine and Cardiology, Medical University of Warsaw, 04-749 Warsaw, Poland; 2School of Public Health, Postgraduate Medical Education Center, 01-813 Warsaw, Poland; 3Department of Physical Education and Health, Faculty in Biala Podlaska, Jozef Pilsudski University of Physical Education in Warsaw, 21-500 Biala Podlaska, Poland; 4Department of Pediatric Cardiology and General Pediatrics, Medical University of Warsaw, 02-091 Warsaw, Poland; 5Students’ Scientific Group of Lifestyle Medicine, 3rd Department of Internal Medicine and Cardiology, Medical University of Warsaw, 04-749 Warsaw, Poland; 6Center for Preclinical Research and Technology CEPT, Department of Experimental and Clinical Pharmacology, Medical University of Warsaw, 02-097 Warsaw, Poland; 7Department of Epidemiology, Cardiovascular Disease Prevention and Health Promotion, National Institute of Cardiology, 04-635 Warsaw, Poland

**Keywords:** COVID-19, endurance athletes, nutrition, diet, physical activity, sport performance, cardiopulmonary exercise test, health promotion, lifestyle interventions, exercise

## Abstract

COVID-19 and imposed restrictions are linked with numerous health consequences, especially among endurance athletes (EA). Unfavorable changes in physical activity and nutrition may affect later sports and competition performance. The aims of this study were: (1) to assess the impact of COVID-19 infection and pandemic restrictions on the nutrition and physical activity of EAs and (2) to compare them with the results of cardiopulmonary exercise testing (CPET). In total, 49 EAs (n_male_ = 43, n_female_ = 6, mean age = 39.9 ± 7.8 year., height = 178.4 ± 6.8 cm, weight = 76.3 ± 10.4 kg; BMI = 24.0 ± 2.6 kg·m^−2^) underwent pre- and post-COVID-19 CPET and fulfilled the dietary and physical activity survey. COVID-19 infection significantly deteriorated CPET performance. There was a reduction in oxygen uptake and in heart rate post-COVID-19 (both *p* < 0.001). Consuming processed meat and replacing meat with plant-based protein affected blood lactate concentration (*p* = 0.035). Fat-free mass was linked with consuming unsaturated fatty acids (*p* = 0.031). Adding salt to meals influenced maximal speed/power (*p* = 0.024) and breathing frequency (*p* = 0.033). Dietary and Fitness Practitioners and Medical Professionals should be aware of possible COVID-19 infection and pandemic consequences among EA. The results of this study are a helpful guideline to properly adjust the treatment, nutrition, and training of EA.

## 1. Introduction

As reported by the World Health Organization, the coronavirus disease-2019 (COVID-19) pandemic has affected the lives of all humanity due to its rapid spread and high mortality rate (https://covid19.who.int (accessed on: 22 November 2022)). Despite the passage of time, a fully effective method of preventing and treating COVID-19 infection has not been found [1]. In addition, a large part of the population remains reluctant to be vaccinated [2], and full vaccination does not completely protect against COVID-19 complications, especially those affecting the circulatory and respiratory systems [3]. A mounting problem for caregivers is the high mutation ability of the virus, which can take various forms [4]. Mutations consist of changes in proteins’ structure and modification of bonds. Due to the unpredictability of the directions of changes, an effective approach to limit spreading and provide sufficient protection in public places is a matter of vital importance.

The period of the pandemic has negatively affected the lifestyle of people around the world. Many people have experienced adverse lifestyle changes [5]. One of the key aspects is changes in physical activity and amount of sitting time [6]. An important aspect related to imposed restrictions is their negative impact on nutrition [7], which may have very serious consequences in the future, such as being overweight, weakening of physical capacity, and civilization diseases [8]. Physical activity and nutrition are both important factors influencing immunity. A potentially better prognosis with COVID-19 infection is achieved by patients consuming more fiber and plants. Such a diet has a positive effect on the gut microflora [9].

The lockdown has affected the performance and lifestyle of professional athletes [10], which may affect later performance in sports. Being ill with COVID-19 has also reduced the number of training days in this population [11], even though athletes are not considered to be at high risk of severe infection and pass it mainly mildly [12]. Studies show that the pandemic has also affected the nutrition of athletes. For example, adverse changes such as increased caloric intake and decreased fruit intake were shown in a group of Rugby players [13]. Moreover, eating disorders associated with mental health deterioration have been noticed among young athletes during the pandemic [14], another study described that the pandemic could have a negative impact on people with eating disorders [15].

COVID-19 disease through multifactorial pathological mechanisms may be accompanied by acute cardiovascular conditions, such as myocarditis, acute myocardial infarction, and cardiomyopathy [16]. The infection may have a long-term negative impact on systemic health status, including the respiratory system due to the risk of respiratory failure, pulmonary thromboembolism, pulmonary embolism, and pneumonia [17]. Due to frequent long-term abnormal lung function, fatigue, shortness of breath, chest pain, and coughing, the disease can be severe for athletes or physically active people [18]. In one study, it was described that post-COVID-19 patients examined by cardiopulmonary exercise testing (CPET) show breathing and decreased peripheral oxygen extraction or decrease in maximum oxygen uptake (VO_2max_) [19].

CPET is a test that can be performed on a treadmill or cycle ergometer and is used to assess aerobic capacity. Is a useful tool to assess progress and plan training [20]. There is a study that shows that the use of CPET as a test to determine persistent symptoms of COVID-19 infection can be helpful in testing endurance athletes [21]. Another study found that patients with persistent dyspnea post-COVID-19 have decreased VO_2max_ as well as present circulatory exercise limitations [22]. An interesting aspect is the influence of nutrition on performance results. For example, a paper comparing a ketogenic diet to a well-balanced diet shows that in the second case even with reduced caloric intake increases in endurance could be achieved [23]. Another study shows that severe deficiencies of micro- and macronutrients such as folic acid and vitamin B12 cause anemia and reduce the performance of endurance work [24]. Thus, paying attention to nutrition among endurance athletes remains a highly important aspect.

We stipulate that previous COVID-19 infections, even with a mild course, paired with imposed restrictions and overall shifts in public life negatively influenced lifestyle and physical performance among endurance athletes. We also hypothesize that those changes are linked together. The aims of this study were: (1) to assess dietary habits, nutrition, and physical activity with their changes caused by mild COVID-19 infection and imposed restrictions, (2) to measure differences in physical performance caused by mild COVID-19 infection, and (3) to compare changes in both areas together and find whether they are correlated.

## 2. Materials and Methods

### 2.1. Study Design

This was a prospective study using data from CPET performed between June 2021 and December 2022. Procedures were conducted in the Tertiary Care Sports Diagnostics Clinic SportsLab (SportsLab, Warsaw, Poland). Participants underwent double CPET pre- and post-COVID-19 infection and had to fulfill a health-assessing questionnaire.

The study group comprised amateur and professional endurance athletes. Inclusion criteria were: (1) CPET performed at the Sportslab Clinic no earlier than 3 years before COVID-19 infection, (2) previous COVID-19 infection (documented by a PCR mRNA test or a positive antigen test) in a period of 14 days to 6 months prior to re-test of CPET, (3) mild course of COVID-19 infection (mandatory no hospitalization), (4) completing each part of the questionnaire, (5) declared lack of actual long-lasting COVID-19 infection consequences, (6) current negative PCR test for COVID-19. The health status of the participants, based on the declared information, was assessed by a medical professional in internal medicine or cardiology before post-COVID-19 CPET. Exclusion criteria were: (1) any respiratory diseases (COPD, poorly controlled bronchial asthma, blood saturation <95%), (2) any cardiovascular diseases (cardiac arrhythmias in the ECG, myocardial ischemia, prolongation of the QT interval in the ECG, structural disorders of the heart in echocardiography, decompensated hypertension with blood pressure >160/100 mmHg), (3) neurological and psychiatric conditions, (4) musculoskeletal disorders, and (5) significant deviations in CBC (leukocytosis >10,000·mm^−3^, anemia with HB levels <10 g·dL^−1^). The flowchart of study procedures is presented in Figure 1.

Each athlete underwent CPET on the same modality pre- and post-COVID-19 infection because each modality was previously linked with different performance results [25]. During the post-COVID-19 CPET, athletes received a questionnaire covering selected areas of general health assessment.

### 2.2. Survey Tool

Subjects fulfilled a questionnaire assessing general health and lifestyle during the CPET post-COVID-19 infection. A validated questionnaire from the PaLS study (Pandemic against LifeStyle project) was used [26,27,28]. The questions covered 3 general areas: demographic information and infection course, diet and eating habits, and physical activity.

Physical activity was rated by the International Physical Activity Questionnaire- Short Form. This part examines three levels of activity: low-, moderate- and vigorous-intensity. Additional questions relate to the period of sitting every day calculated in minutes and hours. Questions related to diet and eating habits were prepared based on Polish National Institute of Public Health recommendations (https://www.pzh.gov.pl/; accessed on 13 November 2022). EAs were asked about the ingredients included in their diet and the intake of particular nutrients (i.e., vegetables and fruits, milk and dairy, processed meat, animal fats and trans fatty acids, sweetened drinks and fruit juices, salt). Moreover, we included questions about eating behaviors (i.e., eating in front of the screen, reading labels of the products, and number of meals pre- and post-COVID-19 pandemic). EAs chose one of the following responses: once a week and less often, 2–3 times a week, most days of the week, every day. “Dietary product” was defined as a food included in the diet (not a dietary supplement) and “Dietary habit” was defined as the habitual decision of individuals regarding what foods they eat. Participants received information about the definition of the above-mentioned terms at the beginning of the survey. Their proper understanding was a mandatory requirement. Demographic and infection-related questions were the author’s questions and assessed, among others, primary sports discipline, sports competition experience, long-lasting COVID-19 consequences, etc.

An additional original question was added to each assessed area, in which respondents rated the impact of the COVID-19 pandemic and previous infection on a given part of their lifestyle or health via a −5/0/+5 scale. Negative (−5/−1) values represented a negative impact, 0 indicated no impact, and +1/+5 values a represented positive impact (values were adjusted for impact strength).

### 2.3. CPET and Somatic Measurements

Body composition was assessed using the Tanita device (Tanita, MC 718, Japan) before each of the double CPETs. Multifrequencies of 5 kHz/50 kHz/250 kHz were used. Body composition analysis and CPET were conducted under the same environmental conditions: ventilated room, an ambient temperature of 20–22 degrees Celsius, and humidity between 40 and 60%. Participants received information on how to properly prepare a few days before the stress test: (1) eat a high-carbohydrate energy meal 2–3 h before, (2) stay hydrated with isotonic sports drinks, (3) avoid intense physical exercise a few days before the test, and (4) be familiar with the test protocol and exercise characteristics on a mechanical treadmill or stationary cycle ergometer.

Cycle CPET was performed on the Cyclus ergometer (RBM elektronik-automation GmbH, Leipzig, Germany). Running CPET was performed on the Cosmos treadmill (h/p/Cosmos quasar, Germany). Cardio-pulmonary measurements were obtained via a Cosmed Quark CPET device (Rome, Italy). Exercises began with a 3–5-min warm-up. It consisted of slow running or walking at a conversational pace (between 7–12 km·h^−1^) or pedaling without resistance. The initial load was individually adjusted for each participant. The speed was then increased every 2 min by 1 km·h^−1^ (constant inclination of 1%), while the load was increased every 2 min by 20 Watts for females and 30 Watts for males. The test was considered terminated if: (1) the participant declared further inability to continue the effort or (2) the VO_2max_ plateau was reached (no increase in the curve of oxygen uptake with increasing exercise intensity). Athletes were encouraged to continue CPET and achieve the best possible result by their supervisor.

The following somatic parameters were measured: weight, height, body fat percentage (BF), fat mass (FM), and fat-free mass (FFM). The following exercise parameters were measured: speed (S) for treadmill or power (P) for cycle ergometry, relative oxygen uptake (VO_2_), absolute oxygen uptake (VO_2a_), heart rate (HR), pulmonary ventilation (VE), breathing frequency (f_R_), and blood lactate concentration (Lac). Each exercise variable was measured at anaerobic threshold (AT), respiratory compensation point (RCP), and maximal exertion. CPET indices were obtained using the breath-by-breath method and a Hans Rudolph V2 Mask (Hans Rudolph Inc, Shawnee, Kansas). VO_2max_ was calculated as the average value of the 10 s period immediately preceding the termination, while HR was considered as the highest value during the stress test. Blood for Lac was collected: (1) before the test, (2) after each increase in intensity, and (3) 3 min after exercise. A 20 μL blood sample was taken each time. The first few drops were drained into a cotton swab before collecting the proper sample. A Super GL2 analyzer (Müller Gerätebau GmbH, Freital, Germany) was used for the analysis, which was calibrated for each participant.

AT was evaluated after meeting the following parameters: (1) the VE/VO_2_ plot was rising with a constant plot of VE/VCO_2_ and (2) the end-tidal partial pressure of oxygen increased at a constant end-tidal partial pressure of carbon dioxide. RCP was evaluated after meeting the following parameters: (1) the VE/VCO_2_ plot had reached its lowest value and started to increase, (2) the decline in the end-tidal partial pressure of carbon dioxide had begun after maximal exertion, (3) there was a rapid increase in VE (2nd deflection), and (4) there was an increase in VCO_2_ relative to VO_2_ (these variables lost their linear relationship) [29].

### 2.4. Ethics

The athletes were tested in accordance with the Declaration of Helsinki. The study protocol was approved by the Bioethics Committee of the Medical University of Warsaw (approval no. KB/50/21 from 19^th^ April 2021). The subjects had to give written consent to participate in the study, published obtained results in the scientific literature and were informed about the potential risks and nuisances related to the procedures. Personal data has been anonymized and does not allow for the identification of the subjects.

### 2.5. Statistical Analysis

Basic data are presented as mean with standard deviation (SD) for continuous variables and numbers (n) with percentage (%) for categorical variables. The data are presented in accordance with APA guidelines (https://apastyle.apa.org/; accessed on 26 November 2022). Normal distribution was assessed using the Shapiro–Wilk test. Between-group differences comparing data from in vivo CPET with declared results from questionnaires were analyzed with Kruskal–Wallis rang’s ANOVA. Differences between CPET and somatic measurements were obtained from Student’s t-test for independent variables.

The total sample size was determined using the G∗Power program (version 3.1.9.2; Düsseldorf, Germany). The significance level was set at *p* = 0.05. The total number of people required is the minimum effective value.

The basic analyses were performed using Excel software (Microsoft Corporation, Washington, USA), and the advanced statistics were conducted in the STATISTICA software (version 13.3, StatSoft Polska Sp. z o.o., Kraków, Poland) and SPSS software (version 28; IBM SPSS, Chicago, IL, USA).

## 3. Results

### 3.1. Basic Population Data

In total, 49 endurance athletes who were referred for CPET and received a survey were included in the study. The baseline description is presented in Table 1. The cohort was 43 (87.8%) males and six (12.2%) females. The average age was 39.9 ± 7.8 years. A total of 31 (62.8 %) endurance athletes completed CPET on the treadmill and 18 (37.7%) on the cycle ergometer. The interval between CPETs was 591.7 ± 282.2 days, while the interval between pre-COVID-19 CPET and negative PCR (indicated as the end of the infection) was 436.4 ± 290.4 and the interval between post-COVID-19 CPET and negative PCR was 155.3 ± 82.52. Twenty (40.8%) athletes declared themselves runners, 14 (28.6%) cyclists, and the remaining 15 (30.6%) declared other disciplines (triathlon, football, and martial arts) as their primary sport. Training experience was 1–2 years for four (8.2%) athletes, 3–5 years for 14 (28.6%) athletes, 6–10 years for 19 (38.8%) athletes and 12 (21.3%) athletes declared >10 years of training experience. COVID-19 infection resulted in waived competition for 23 (46.9%) participants. On a 0–5 scale subjects declared their overall health as 4.8 ± 0.5 before infection and as 4.1 ± 0.5 after. Ten (20.4%) participants suffered from long-lasting COVID-19 consequences for >14 days.

### 3.2. CPET Characteristics

We found significant differences between CPET performance conducted pre- and post- \COVID-19 infection. The most significant differences were noted for VO_2_ at AT, RCP, and maximum (each *p* < 0.001). The mean VO_2_ before infection was 35.0 ± 6.4 mL·kg·min^−1^, 43.9 ± 7.3 mL·kg·min^−1^, and 47.8 ± 7.8 mL·kg·min^−1^, respectively, for AT, RCP, and maximal. Meanwhile, post-infection values were 32.4 ± 5.9 mL·kg·min^−1^, 40.5 ± 6.6 mL·kg·min^−1^, and 45.0 ± 7.0 mL·kg·min^−1^, respectively, for AT, RCP, and maximal. In HR measurements there was also an aggravation at AT and RCP (both *p* < 0.001). The peak HR before infection was 145.1 ± 10.8 bpm for AT and 168.8 ± 9.0 bpm for RCP. Post-infection values were 141.1 ± 10.0 bpm for AT and 165.1 ± 9.7 bpm for RCP. Lac_max_ and VE only deteriorated for RCP. Their precise values are presented in Table 2.

### 3.3. Nutrition and Eating Habits

Descriptive results of the survey along with mean rangs have been presented in Table 3. Participants received information about the definition of “Dietary product” and “Dietary habit” at the beginning of the survey. Due to a high number of possible combinations, only significant results (with *p* < 0.05) have been shown. A Kruskal–Wallis H-test showed significant differences between the frequency of particular dietary products, eating habits, and CPET scores. Precise results are presented in Table 4. Consuming products containing processed meat significantly affected Lac_max_ [H(3) = 8.9; *p* = 0.030]. The highest Lac_max_ was noted for participants who chose “Majority of days within a week” compared to other responses (mean rang= 20.1). However, replacing meat with protein-rich plant products such as nuts and legumes influenced both Lac_max_ [H(3) = 8.6; *p* = 0.035] and maximal HR [H(3) = 8.6; *p* = 0.036]. Athletes who again selected “Majority of days within a week” had the highest Lac_max_ (mean rang= 19.2) and HR_max_ (mean rang= 32.4). It is worth mentioning that FFM was linked with consuming products that are sources of unsaturated fatty acids [H(3) = 8.9; *p* = 0.031]. Adding salt to meals influenced maximal speed or power [H(3) = 9.4; *p* = 0.024] and f_R_ at RCP [H(3) = 13.7; *p* = 0.033].

### 3.4. Physical Activity

Declared results of physical activity questions are presented in Table 5. There were not any significant differences (each *p* > 0.05) between the frequency of selected answers in the International Physical Activity Questionnaire—Short Form and CPET performance.

## 4. Discussion

The detrimental effect of COVID-19 infection on nutrition and eating habits and CPET results in the group of endurance athletes was shown. The main findings were: (1) among changes in eating habits, the most commonly reported were drinking sweetened drinks and fruit juices instead of water, more salting of food, and consumption of more meals per day compared to the pre-pandemic time; (2) post-COVID-19 status influenced outcomes of CPET such as VO_2_ at AT, RCP, and maximum, as well as HR at AT and RCP (each *p* <0.001); (3) eating habits during the pandemic influenced the achieved results of CPET, e.g., Lac_max_ increased in people who more often consumed products containing processed meat and Lac_max_ and HR_max_ were affected by replacing meat with protein-rich plant products such as nuts and legumes.

There are studies describing the impact of COVID-19 on CPET scores and the physical performance of patients. One study shows that post-COVID-19 people had a reduced VO_2_ score compared to the control group, and they also showed more respiratory failure (VE/VCO_2_) [30]. This means that this study, like our study, did not achieve the expected result at peak exercise, which may be due to the long-term effects of the disease on the cardiovascular or respiratory systems [31]. In a study in a sample of elite swimmers that compared CPET results in athletes who had the infection and athletes who did not have the disease, no evidence of the effect of the disease on the variability of results was found [32]. A similar study on volleyball players showed results typical of detraining, but without access to the patient’s results before the onset of the disease, it is difficult to assess [33]. The conclusions differ from our results, but the reason for this may be the different course of the disease and its impact on healthy people, especially athletes who are not considered to be at risk [34]. The next study assessed the exercise capacity of mildly symptomatic patients using an ergometer, comparing the results before and after contracting COVID-19. Slight differences in MET and VO_2peak_ results were noted [35]; median VO_2max_ in the cited study was 21,857 pre-COVID-19 and 21,699 post-COVID-19, while in our study pre-COVID-19 it was 3623.47 and post-COVID-19, 3406.00 (ml·kg·min^−1^), which proves that athletes, despite the disease, have a greater capacity than other patients. Another study claims that symptomatic patients with the post-COVID-19 syndrome were more likely to show lower main VO_2peak_, less likely to reach the AT, and symptoms of the syndrome worsened during examinations [36]. In patients with long-term symptoms after infection, 55% had reduced pVO_2peak_ in CPET, and 31% had reduced pVO_2peak_ secondary to limitation and muscular impairment [37]. The data from these studies are worrying because they show the long-term impact on patients’ health. Reduced performance deteriorates the quality of life of any patient, especially professional athletes. It was recognized that VO_2max_ could be used as a clinical tool for estimating the severity of COVID-19 infection [38]. It turns out that the VO_2_ result can also be a good predictor of mortality in patients with SARS-CoV-2 [37]; therefore, the use of the CPET test in our work reliably showed the impact of COVID-19 on the health and performance of patients. The probable explanation why we observed the decrease in HR among the EAs may be due to age-related changes in the cardiovascular system [39]. Although some time had elapsed between both CPETs, time-related impact is justified. It is worth noting here that Komici et al. observed a change in forced expiratory volume in one second examined in spirometry in EAs who had survived COVID-19 [40].

There is no doubt that proper nutrition is one of the most important components of human health and life. Professional athletes in their daily diet care for the building of muscles, but also for the proper development of other tissues (such as the vascular system and central nervous system) [41] and providing the right amount of energy to the body. Adequate nutrition for endurance athletes is difficult because it depends on many factors, such as the amount and type of training, time to competition, and type of exercise performed [42]. Even more so, the period of the COVID-19 pandemic and disease did not contribute positively to the improvement of nutrition [28]. Evidence supports that high carbohydrate intake provides muscle glycogen stores and prevents hypoglycemia [43], and there is debate about fluid intake as water poisoning has occurred in the ultra-resistant environment [44].

We recommend further similar studies that include analysis adjusted with participants’ competition results. It is worth mentioning that one study conducted on Olympic athletes did not affect the results achieved during the competition [45], but more research should be performed to deeply investigate those findings. Many competitions were canceled during the pandemic and EAs had no chance to face them. Returning to competition after a long break should be well thought out [46]. Moreover, other competitions had to be adapted to the new circumstances. In the case of a long-term break from training, it is worth using rehabilitation aimed at improving respiratory efficiency. In addition to physical preparation, mental and dietary preparation are also important [47,48]. Thus, EAs during and after illness require special attention to their mood and nutrition.

Our respondents found that more frequent consumption of processed meat increases Lac_max_, which is not desired by professional athletes [49]. Excessive consumption of processed meat can even be carcinogenic [50]. As described in the Results, it was pointed out that up to 59.2% of participants eat processed meat more than two times per week. In a study that investigated the effect of a plant-based diet on cardiovascular disease, it turned out to be positive because it lowered blood pressure, HR, and insulin levels [51]. Furthermore, our respondents, who were more likely to use plant-based protein products, had an aggravation in HR and Lac_max_. Among studies examining the effect of unsaturated fats on health, one of them showed that the consumption of such fats is associated with a reduction in the risk of cardiovascular disease, and also increases the share of lean body mass [52], and this was also achieved by the athletes in our survey. Another study in mice showed that eating unsaturated fats compared to saturated fats provides more energy and endurance [53]. Excessive salt added to food can have long-term effects such as the risk of high blood pressure, stroke, kidney, and heart disease [54]. There are studies that show health problems in athletes who consume too little and too much sodium, and salt should be replaced after exercise, remembering to hydrate reasonably [55]. These and other aspects of nutrition are very important for professional athletes due to their impact on body structure and endurance, so it is important to take care of meals even in the case of illness and the COVID-19 pandemic. A proper diet introduced after recovering from COVID-19 can reduce the long-term effects of the disease [56]. However, the explanation of the physiological basis of the obtained results is beyond the scope of this paper. Thus, we recommend it for further research.

### Limitations

Probably due to the small sample size and structure of questions, we did not observe any significance in the results of the physical activity questionnaire. The limitation of our questionnaire is the possibility that the physical activity and nutrition of the respondents changed during the pandemic; in addition, they may not remember their old habits. There was a relatively long period of time between the two CPET tests, so this may have influenced the changes in parameters. We are aware of the existing limitations, therefore we recommend carefully evaluating the results.

## 5. Conclusions

EAs require special professional care to be ready for sports competitions. Training loads and nutrition status, especially considering the period after infection, should be carefully adjusted. The results of this study may be used in determining the training and nutrition plans of professional athletes. Nutritionists should consider what eating habits emerge when athletes have limited training opportunities and spend more time at home. It is important to remember the adequate supply of fluids, fruits and vegetables, vitamins, and whole grains. Thanks to these results, doctors and trainers will be more aware of how performance and the body changes after suffering from COVID-19. Gradual adaptation of EAs to loads after a break in training is crucial to staying in shape and avoiding injury. Rehabilitation may also be required in the event of greater damage to health and condition. This will allow them to choose the right treatment and tests to avoid long-term complications. Each EA should be treated individually and comprehensive care should be implemented to facilitate recovery.

## Figures and Tables

**Figure 1 nutrients-14-05381-f001:**
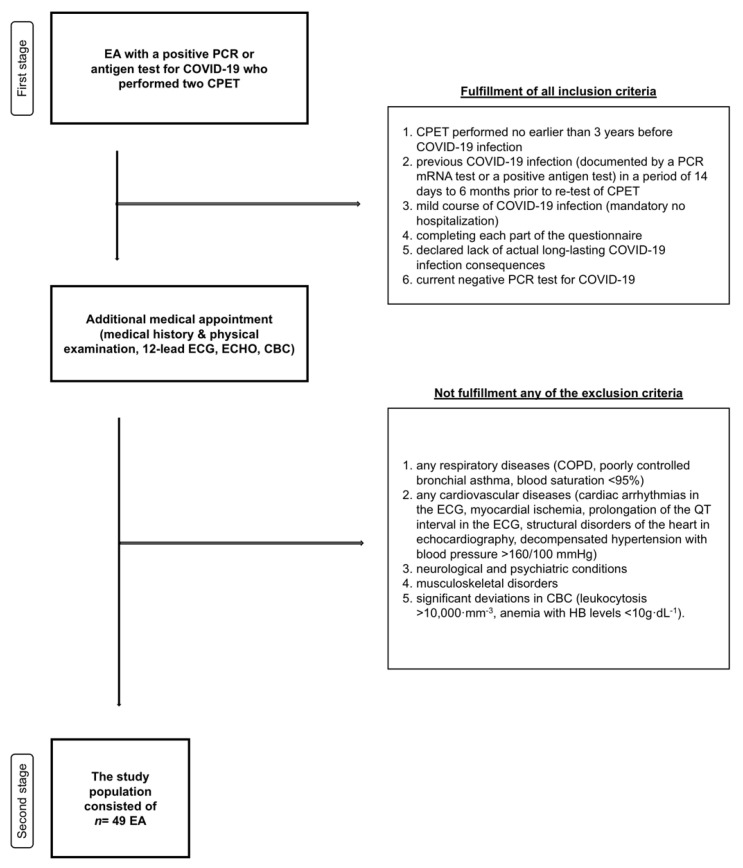
Study procedure. Abbreviations: EA, endurance athlete; PCR, polymerase chain reaction; COVID-19, coronavirus disease 2019; CPET, cardiopulmonary exercise test; ECG, 12-lead electrocardiogram; ECHO, echocardiography examination; CBC, complete blood count; COPD, Chronic obstructive pulmonary disease.

**Table 1 nutrients-14-05381-t001:** Participants’ characteristics.

Variable		Males (*n* = 43)	Females (*n* = 6)
Age (years)		40.7 ± 7.0	38.1 ± 6.4
Height (cm)		178.5 ± 6.8	178.4 ± 6.9
CPET modality	Treadmill	25 (51.0%)	4 (8.2%)
	Cycle ergometer	16 (32.7%)	2 (4.1%)
Primary sport discipline	Running	16 (32.7%)	4 (8.2%)
	Cycling	13 (26.5%)	1 (2.0%)
	Other	14 (28.6%)	1 (2.0%)
Waived competition due to COVID-19 infection	Yes	21 (42.9%)	2 (4.1%)
	No	23 (46.9%)	4 (8.2%)
	Pre-COVID-19	Post-COVID-19	*p*-value
Weight (kg)	76.6 ± 10.0	76.7 ± 10.9	0.951
BMI (kg·m^−2^)	24.0 ± 2.5	24.0 ± 2.7	0.931
FFM (kg)	63.4 ± 7.6	63.5 ± 8.0	0.774
BF (%)	17.1 ± 4.7	16.9 ± 5.1	0.604
FATM (kg)	13.3 ± 4.7	13.2 ± 5.2	0.848

Abbreviations: COVID-19, coronavirus disease 2019; BMI, body mass index; FFM, fat-free mass; BF, body fat; FATM, fat mass. Data are presented as means with standard derivations (±) for continuous variables and numbers with percentages (%) for categorical variables.

**Table 2 nutrients-14-05381-t002:** Differences in CPET performance before and after COVID-19 infection.

Variable	Pre-COVID-19	Post-COVID-19	*p*-Value
VO_2AT_ (mL·kg·min^−1^)	35.0 ± 6.5	32.4 ± 6.0	<0.001
VO_2Ata_ (mL·min^−1^)	2650.0 ± 470.9	2446.1 ± 400.3	<0.001
HR_AT_ (beats·min^−1^)	145.1 ± 10.9	141.1 ± 10.1	0.001
VE_AT_ (L·min^−1^)	70.8 ± 18.7	68.1 ± 14.7	0.090
S_AT_ (km·h^−1^)	11.4 ± 1.4	11.1 ± 1.3	0.044
P_AT_ (Watts)	162.8 ± 25.9	154.8 ± 25.9	0.066
f_RAT_ (breaths·min^−1^)	32.1 ± 9.0	32.1 ± 8.1	0.706
Lac_AT_ (mmol·L^−1^)	2.0 ± 0.9	2.1 ± 0.9	0.630
VO_2RCP_ (mL·kg·min^−1^)	43.9 ± 7.4	40.5 ± 6.7	<0.001
VO_2RCPa_ (mL·min^−1^)	3324.3 ± 512.9	3063.7 ± 440.1	<0.001
HR_RCP_ (beats·min^−1^)	168.8 ± 9.2	165.1 ± 9.8	<0.001
VE_RCP_ (L·min^−1^)	106.8 ± 21.7	98.9 ± 18.3	<0.001
S_RCP_ (km·h^−1^)	14.3 ± 1.9	13.8 ± 1.5	<0.001
P_RCP_ (Watts)	245.2 ± 42.0	232.2 ± 39.7	0.061
f_RRCP_ (breaths·min^−1^)	41.3 ± 8.7	40.1 ± 8.9	0.876
Lac_RCP_ (mmol·L^−1^)	4.9 ± 1.4	4.3 ± 1.1	0.013
VO_2max_ (mL·kg·min^−1^)	47.8 ± 8.0	45.0 ± 7.1	<0.001
VO_2maxa_ (mL·min^−1^)	3623.5 ± 552.1	3406.0 ± 474.5	<0.001
HR_max_ (beats·min^−1^)	180.8 ± 10.1	179.8 ± 10.0	0.273
VE_max_ (L·min^−1^)	143.0 ± 26.9	138.50 ± 23.9	0.068
S_max_ (km·h^−1^)	16.6 ± 1.6	16.4 ± 1.7	0.264
P_max_ (Watts)	310.0 ± 37.2	312.2 ± 49.1	0.811
f_Rmax_ (breaths·min^−1^)	58.9 ± 14.4	57.3 ± 11.0	0.959
Lac_max_ (mmol·L^−1^)	9.7 ± 2.3	9.6 ± 2.4	0.880

Abbreviations: COVID-19, coronavirus disease 2019; VO_2AT_, oxygen uptake at the anaerobic threshold; VO_2ATa_, absolute oxygen uptake at the anaerobic threshold; HR_AT_, heart rate at the anaerobic threshold; VE_AT_, pulmonary ventilation at the anaerobic threshold; S_AT_, speed at the anaerobic threshold; P_AT_, power at the anaerobic threshold; f_RAT_, breathing frequency at the anaerobic threshold; VO_2RCP_, oxygen uptake at the respiratory compensation point; VO_2RCPa_ absolute oxygen uptake at the respiratory compensation point; HR_RCP_, heart rate at the respiratory compensation point; VE_RCP_ pulmonary ventilation at the respiratory compensation point; S_RCP_, speed at the respiratory compensation point; P_AT_, power at the respiratory compensation point; f_RRCP_, breathing frequency at the respiratory compensation point; Lac_RCP_, blood lactate concentration at the respiratory compensation point; VO_2max_, maximal oxygen uptake; VO_2maxa_, absolute maximal oxygen uptake; HR_max_, maximal heart rate; VE_max_, maximal pulmonary ventilation; S_max_, maximal speed, P_max_, maximal power; f_Rmax_, maximal breathing frequency; Lac_max_, maximal blood lactate concentration. Data are presented as means with standard derivations (±). Speed is presented for treadmill CPET (*n* = 29) and power is presented for cycle ergometer CPET (*n* = 18).

**Table 3 nutrients-14-05381-t003:** Nutrition and dietary habits of participants.

Question	Frequency				Lack of Answer
	Once a Week or Less Often	2–3 Times per Week	Majority of Days Within a Week	Every Day	
	*N* (%)	*N* (%)	*N* (%)	*N* (%)	*N* (%)
Consuming more meals per day than before the pandemic (including snacking)	32 (65.3%)	7 (14.3%)	7 (14.3%)	0 (0.0%)	3 (6.1%)
Consuming less than 3 servings of wholegrain products daily (less than 90 g/day)	20 (40.8%)	18 (36.7%)	6 (12.2%)	2 (4.1%)	3 (6.1%)
Consuming less than 400 g vegetables and fruits	22 (44.9)	11 (22.5%)	11 (22.5%)	2 (4.1%)	3 (6.122%)
Consuming less than 2 glasses of unsweetened milk or other dairy products daily	24 (49.0%)	9 (18.4%)	8 (16.3%)	5 (10.2%)	3 (6.1%)
Consuming products containing processed meat, (such as sausages, ham, frankfurters, etc.)	20 (40.8%) 10.9 for Lac_max_	15 (30.6%) 16.6 for Lac_max_	8 (16.3%)20.1 for Lac_max_	2 (4.1%)4.5 for Lac_max_	4 (8.2%)
Replacing meat with protein-rich plant products such as nuts and legumes: beans, chickpeas, soy, lentils, fava beans, peas	17 (34.7%) 19.1 for HR_max_ 10.9 for Lac_max_	14 (28.6%) 27.3 for HR_max_ 18.1 for Lac_max_	8 (16.3%)32.4 for HR_max_19.2 for Lac_max_	7 (14.3%)16.3 for HR_max_7.1 for Lac_max_	3 (6.1%)
Consuming products that aresources of animal fats or trans fatty acids present in products, such as pastries, candy bars, salty snacks, and fast-food products	24 (49.00%)	12 (24.5%)	7 (14.3%)	2 (4.1%)	4 (8.2%)
Consuming products that are sources of unsaturated fatty acids, such as canola oil, olive oil, or fish	9 (18.4%) 35.2 for FFM	20 (40.8%) 19.6 for FFM	11 (22.5%)21.1 for FFM	6 (12.2%)23.4 for FFM	3 (6.1%)
Drinking sweetened beverages or fruit juices instead of water	40 (81.6%)	5 (10.2%)	2 (4.1%)	0 (0.0%)	2 (4.1%)
Adding salt to meals	27 (55.1%) 26.7 for S_max_/P_max_ 21.0 for f_RRCP_	9 (18.4%) 25.2 for S_max_/P_max_ 37.7 for f_RRCP_	6 (12.2%)17.5 for S_max_/P_max_20.5 for f_RRCP_	4 (8.2%)7.1 for S_max_/P_max_13.0 for f_RRCP_	3 (6.1%)
Consuming meals while looking at the screen of a TV, computer, or other device	21 (32.9%)	12 (24.5%)	9 (18.4%)	4 (8.2%)	3 (6.1%)
Paying attention to labels of chosen products during shopping, taking into account ingredients, amount of calories, etc.	10 (20.4%)	6 (12.2%)	11 (22.4%)	19 (38.8%)	3 (6.1%)
Self-assessed impact of COVID-19 pandemic and imposed restrictions on nutrition and dietary habits [in −5/0/+5 scale]	0.09 ± 1.43				0 (0.0%)

Abbreviations: COVID-19, coronavirus disease 2019; na, Lac_max_, maximal blood lactate concentration; HR_max_, maximal heart rate; f_RRCP_, breathing frequency at the respiratory compensation point; S_max_, maximal speed; P_max_, maximal power. Data are presented as numbers (N) with percentages (%) of the whole population and in the −5/0/+5 scale question data are presented as means with standard deviations (±). The mean rang has been added only for significant differences (*p* < 0.05) between a particular group and CPET performance. The mean rang was calculated by the Kruskal–Wallis H-test.

**Table 4 nutrients-14-05381-t004:** Relationships between nutrition, dietary habits, and CPET performance.

CPET Variable	Survey Question	*p*-Value
Lac_max_	How often did you eat processed meat products?	0.030
HR_max_	How often did you replace meat with plant-based protein products?	0.036
Lac_max_	How often did you replace meat with plant-based protein products?	0.035
FFM	How often unsaturated fats were the main source of fat in your diet?	0.031
S_max_/P_max_	How often did you add salt to your meals?	0.024
f_RRCP_	How often did you add salt to your meals?	0.033

Abbreviations: Lac_max_, maximal blood lactate concentration; HR_max_, maximal heart rate; FFM, fat-free mass; S_max_, maximal speed; P_max_, maximal power; f_RRCP_, breathing frequency at the respiratory compensation point. Due to a high number of possible combinations, only significant results (with *p* < 0.05) have been shown.

**Table 5 nutrients-14-05381-t005:** Results of physical activity questions.

Type of Activity	Amount	Median (IQR)
Vigorous physical activity	Days/week	3 (2.8–5)
	Minutes/week	60 (43.8–90)
Moderate physical activity	Days/week	3 (2–4.3)
	Minutes/week	50 (30–90)
Walking	Days/week	6.5 (4–7)
	Minutes/week	40 (20–60)
Sitting time	Hours/day	8 (5–9.3)
Weekly running distance (question only for runners)		25 (0–50)
Weekly cycling distance (question for cyclists)		70 (3.75–200)
Self-assessed fitness level in pre-COVID-19 [in 0–10 scale]		8 (6–9)
Self-assessed fitness level post-COVID-19 [in 0–10 scale]		6 (4.8–8)
Self-assessed impact of COVID-19 pandemic and imposed restrictions on fitness level [in −5/0/+5 scale]		−1 (−3–0)

Abbreviations: IQR, interquartile range; COVID-19, coronavirus disease 2019. Data are presented as medians with the interquartile range due to their non-normal distribution.

## Data Availability

The data presented in this study are available on request from the corresponding author. The data are not publicly available due to not obtaining consent from respondents to publish the data.

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
