# Peer review of "The Influence of Nutrition and Physical Activity on Exercise Performance after Mild COVID-19 Infection in Endurance Athletes-CESAR Study"

_nutrients, 2022, doi:10.3390/nu14245381_

Round 1

Reviewer 1 Report

The manuscript by Śliż et al. “The influence of nutrition and physical activity on exercise performance after mild COVID-19 infection in endurance athletes CESAR Study” depicted the consequences assessment of COVID-19 infection and pandemic restrictions on nutrition and physical activity, especially among endurance athletes and its validation via cardiopulmonary exercise testing. Overall, the manuscript is fascinating and requires major revision to justify its proceed.

Comments

1.      Please add the illustration of strategy and outcomes to be explained by illustrations.

2.      The diet and gut microbiota have potentially helpful in maintaining immunity and health against COVID-19. Please add some information on its mechanism in the Introduction section, i.e., Indian Journal of Microbiology 60 (2020) 420-429.

3.      In the present scenario, we are still facing various therapeutic challenges against COVID-19 and the emergence of its variants evolution. Please, highlight such information with a few examples, mechanisms of their evolution, and challenges for potential threats of COVID-19 variants i.e, Infection 50 (2022) 309-325.

4.      Lines 85-92 clearly present the details of the investigation's primary objectives, novelty, and significance.

5.      Discussion section is weak; this section can be adequately explained with quantitative literature data and an in-depth mechanism to justify the significance of the finding.

6.      Line 351, please also add some information on potential challenges and perspectives.

7.      Please avoid the use of citations in the conclusions section.

Author Response

Warsaw, December 8th, 2022

Dear Reviewer,

On behalf of co-authors of the paper entitled: “The influence of nutrition and physical activity on exercise performance after mild COVID-19 infection in endurance athletes- CESAR Study” we would like to thank you for your time and contribution while reviewing our manuscript. We did our best to revise our paper in accordance with your suggestions. In this file, you can find our responses. We uploaded a revised manuscript in a “tracking changes form” in MDPI Submission System.

We hope that all points were properly addressed and that our paper in its present form will fulfil the requirements for publication in the Nutrients.

Commentary: Please add the illustration of strategy and outcomes to be explained by illustrations.

Response: Thank you for your valuable suggestion. We add Figure 1 which presented the flowchart of the study.

Commentary: The diet and gut microbiota have potentially helpful in maintaining immunity and health against COVID-19. Please add some information on its mechanism in the Introduction section, i.e., Indian Journal of Microbiology 60 (2020) 420-429.

Response: Thank you for suggesting this literature. We used it in the introduction section and it enriched our introduction with new, interesting information.

Commentary: In the present scenario, we are still facing various therapeutic challenges against COVID-19 and the emergence of its variants evolution. Please, highlight such information with a few examples, mechanisms of their evolution, and challenges for potential threats of COVID-19 variants i.e, Infection 50 (2022) 309-325.

Response: We added the mentioned issues to the introduction and quoted the study. We developed the issues of mechanisms and infection prevention, which ties in with the previous point.

Commentary: The discussion section is weak; this section can be adequately explained with quantitative literature data and an in-depth mechanism to justify the significance of the finding.

Response: According to the comment, we enriched the discussion with further observations and citations. We expanded the issue of diet and the impact of variables on athletic performance. We also discussed the issue of resignation from the competition and their cancellations due to the pandemic.

Commentary: Line 351, please also add some information on potential challenges and perspectives.

Response: As suggested, we have extended the guidance and future application of the study in the conclusions section. It was a valuable suggestion.

Commentary: Please avoid the use of citations in the conclusions section.

Response: Thank you for this note, there is no more citation in this section, we have moved it to another part.

To sum up all the above answers, we again thank you for your precise review. If you have more comments, do not hesitate to contact us.

Reviewer 2 Report

Journal           Nutrients (ISSN 2072-6643)

Manuscript ID         nutrients-2101504

Type  Article

Title       The influence of nutrition and physical activity on exercise performance after mild COVID-19 infection in endurance athletes- CESAR Study

Dear authors:

A good scientific work of the publication is presented.

The title is good and quite specific and attracts the reader's interest so that it is easy to understand.

The topic of scientific research is quite interesting and relevant.

The scientific article is logically built, and corresponds to the principles of presenting scientific information and research.

I think that the article used enough tables and illustrations.

There are some minor suggestions:

Abstract:

I suggest the authors improve. This should be done in order to attract more attention of readers to research on this scientific topic.

keywords

I suggest adding a few terms to expand the search (this is a recommendation)

Introduction

I suggest that the authors significantly improve, expand the search for scientific articles on this topic, perform a comprehensive analysis, and focus on the problems of the scientific direction and its significance for further research.

Materials and Methods

It is in this section that I recommend describing the characteristics of the study participants (number, gender, age, and other characteristics).

The authors did not indicate how the volunteers for the scientific experiment were selected.

It is in this section that it is necessary to indicate in more detail the main questions of the questionnaire, and not just the sections of the questionnaire.

Was the consent of the respondents obtained for the publication of the obtained data in open sources of information?

3. Results

dietary products – it is necessary to give an explanation of the term when analyzing the eating behavior of the surveyed athletes.

dietary habits of participants - the level of understanding of the given term by the surveyed athletes

4. Discussion

I recommend improving this section, and structuring it with your research findings.

Our respondents ………… excessive consumption of processed meat can even be carcinogenic - please describe and rate the consumption of meat products.

Why is an important aspect of the body's supply of minerals, essential amino acids, vitamins, and so on not touched upon?

Conclusions

It is necessary to significantly rework, supplement, and expand in accordance with the purpose and objectives of the research.

References

I believe that the authors of the article approached the analysis of the problem with sufficient care and used the required number of citations.

42 sources of scientific information (84 percent) presented in the List of References (bibliographic list) have been published over the past 5 years.

I draw your attention to the style of registration in this section, a correction is needed. This source of information should preferably be placed in the text of the article https://covid19.who.int (accessed on: 22nd November 2022) and removed from the general list (References)

I recommend for publication the scientific article "The influence of nutrition and physical activity on exercise performance after mild COVID-19 infection in endurance athletes- CESAR Study" - nutrients-2101504, subject to minor improvements and adjustments.

Prof. Dr. Maksim Rebezov

V. M. Gorbatov Federal Research Center for Food Systems of Russian Academy of Sciences, Moscow, Russian Federation

  Максим Борисович Ребезов            10.12.2022

E-mail: rebezov@ya.ru

https://www.scopus.com/authid/detail.uri?authorId=57507188200

https://www.webofscience.com/wos/author/record/E-5487-2016

https://elibrary.ru/author_profile.asp?id=419764

http://orcid.org/0000-0003-0857-5143

https://www.researchgate.net/profile/Maksim-Rebezov

Author Response

Warsaw, December 11th, 2022

Dear Reviewer,

On behalf of the co-authors of the paper entitled: “The influence of nutrition and physical activity on exercise performance after mild COVID-19 infection in endurance athletes- CESAR Study” we would like to thank you for your time and contribution while reviewing our manuscript. We did our best to revise our paper in accordance with your suggestions. In this file, you can find our responses. We uploaded a revised manuscript in a “tracking changes form” in MDPI Submission System.

We hope that all points were properly addressed and that our paper in its present form will fulfil the requirements for publication in the Nutrients.

Commentary:  

Abstract:

I suggest the authors to improve. This should be done in order to attract more attention of readers to research on this scientific topic.

Response: Thank you for the suggestion. Although, we received feedback about our abstract from the remaining two Reviewers and they suggested that it is suitable, attractive, and concise.

Commentary:  

keywords

I suggest adding a few terms to expand the search (this is a recommendation)

Response: Thank you for your valuable suggestion. We add some new keywords. That’s improved this section.

Commentary:

Introduction

I suggest that the authors significantly improve, expand the search for scientific articles on this topic, perform a comprehensive analysis, focus on the problems of the scientific direction and its significance for further research.

Response: We revised our Introduction. Major improvements (marked with tracking changes) and additional minor corrections have been added. Moreover, we enrich our Introduction with a new body of literature.

 Commentary:  

Materials and Methods

It is in this section that I recommend to describe the characteristics of the study participants (number, gender, age, and other characteristics).

The authors did not indicate how the volunteers for the scientific experiment were selected.

It is in this section that it is necessary to indicate in more detail the main questions of the questionnaire, and not just the sections of the questionnaire.

Was the consent of the respondents obtained for the publication of the obtained data in open sources of information?

Response: Thank you for each suggestion related to Materials and Methods.

We prefer to present population characteristics in the Results paragraph along with other findings. This approach enables us to present all study outcomes in one section which is more concise and suitable for the reader.

We add Figure 1 which presents the whole study selection procedures.

We precisely described all points of our survey.

We received the informed consent of each EAs to publish obtained findings in scientific literature. The document described all study purposes and procedures. Providing assigned paper was a mandatory requirement for participants.

Commentary:  

  1. Results 

dietary products – it is necessary to give an explanation of the term when analyzing the eating behavior of the surveyed athletes.

dietary habits of participants - the level of understanding of the given term by the surveyed athletes

Response: Our participants received information about definitions of the above-mentioned terms at the beginning of the questionnaire. We underlined this fact in the Results section (see paragraph 3.3). Understanding those terms was a mandatory inclusion requirement. We underlined how the dietary habit and dietary products were described in the Materials and Methods (see 2.2 paragraph).

Commentary:  

  1. Discussion

I recommend improving this section, structuring it with your research findings.

Our respondents ………… excessive consumption of processed meat can even be carcinogenic - please describe and rate the consumption of meat products.

Why is an important aspect of the body's supply of minerals, essential amino acids, vitamins, and so on not touched upon?

Response: Thank you for both suggestions. We described reasons for the importance of a proper supply of macronutrients supported by the body of literature. Moreover, we provided the rate of meat consumption obtained in our survey.

Commentary:  

Conclusions

It is necessary to significantly rework, supplement, expand in accordance with the purpose and objectives of the research.

Response: Thank you for the valuable commentary. We rewrote the conclusions. Now, this paragraph is more concise and better points out the importance of the purpose and objectives of our research.

Commentary:  

References

I believe that the authors of the article approached the analysis of the problem with sufficient care and used the required number of citations.

42 sources of scientific information (84 percent) presented in the List of References (bibliographic list) have been published over the past 5 years.

I draw your attention to the style of registration in this section, a correction is needed. This source of information should preferably be placed in the text of the article https://covid19.who.int (accessed on: 22nd November 2022) and removed from the general list (References)

Response: We add the new body of references (including newly published papers) when we revised our Introduction and Discussion sections. New citations improved the quality of our paper. Moreover, we transferred Reference No. 1- https://covid19.who.int (accessed on: 22nd November 2022) to the text.

To sum up all the above answers, we again thank you for your precise review. If you have more comments, do not hesitate to contact us.

Reviewer 3 Report

The paper can be proceed after a short revision. Two points should be addressed:

1) the Authors claim that an increase in the heart rate was observed in the group of post-Covid athletes, However, the numbers presented in the Table 2 are contraditory: the rate decreases in the post-Covid group both at the anaerobic threshold and at the respiratory compensation point. Interestingly, the maximal rate remains unchanged, despite decreased oxygen uptake (Table 2). How can the Authors explain such a discrepancy ?

2) Are the changes in the CPET performance, such as decreased oxygen uptake , related to putatively impaired sports performance in the group of post-Covid athletes ? Please, share your ideas or data in this field.

Author Response

Warsaw, December 8th, 2022

Dear Reviewer,

On behalf of co-authors of the paper entitled: “The influence of nutrition and physical activity on exercise performance after mild COVID-19 infection in endurance athletes- CESAR Study” we would like to thank you for your time and contribution during reviewing our manuscript. We did our best to revise our paper in accordance with your suggestions. In this file, you can find our responses. We uploaded a revised manuscript in a “tracking changes form” in MDPI Submission System.

We hope that all points were properly addressed and that our paper in its present form will fulfill the requirements for publication in the Nutrients.

Commentary: the Authors claim that an increase in the heart rate was observed in the group of post-Covid athletes, However, the numbers presented in the Table 2 are contraditory: the rate decreases in the post-Covid group both at the anaerobic threshold and at the respiratory compensation point.

Response: Thank you very much for this request. The inaccuracy resulted from our oversight, we corrected this sentence in the discussion and now it agrees with the result.

Commentary: Interestingly, the maximal rate remains unchanged, despite decreased oxygen uptake (Table 2). How can the Authors explain such a discrepancy ?

Response: Thank you for this valuable suggestion about research inaccuracies. The most physiologically likely response is a decline in HR with age. We have included the relevant sentence and quote in the discussion.

Commentary: Are the changes in the CPET performance, such as decreased oxygen uptake , related to putatively impaired sports performance in the group of post-Covid athletes ? Please, share your ideas or data in this field.

Response: We do not have information on the results achieved by our endurance athletes, some of them had to withdraw from the competition due to infection (Table 1). A lack of information on sports results has been placed in the discussion, we reported another study that covers this topic.

To sum up all the above answers, we again thank you for your precise review. If you have more comments, do not hesitate to contact us.

Round 2

Reviewer 1 Report

Accept as is